

# Spatial temporal distribution of COVID-19 risk during the early phase of the pandemic in Malawi

Alfred Ngwira, Felix Kumwenda, Eddons C.S. Munthali and Duncan Nkolokosa

Basic Sciences Department, Lilongwe University of Agriculture and Natural Resources, Lilongwe, Malawi

## ABSTRACT

**Background:** COVID-19 has been one of the greatest challenges the world has faced since the second world war. This study aimed at investigating the distribution of COVID-19 in both space and time in Malawi.

**Methods:** The study used publicly available data of COVID-19 cases for the period from 2 April 2020 to 28 October 2020. Semiparametric spatial temporal models were fitted to the number of monthly confirmed cases as an outcome data, with time and district as independent variables, where district was the spatial unit, while accounting for sociodemographic factors.

**Results:** The study found significant effects of location and time, with the two interacting. The spatial distribution of COVID-19 risk showed major cities being at greater risk than rural areas. Over time, the COVID-19 risk was increasing then decreasing in most districts with the rural districts being consistently at lower risk. High proportion of elderly people was positively associated with COVID-19 risk ($\beta$ = 1.272, 95% CI [0.171, 2.370]) than low proportion of elderly people. There was negative association between poverty incidence and COVID-19 risk ($\beta$ = −0.100, 95% CI [−0.136, −0.065]).

**Conclusion:** Future or present strategies to limit the spread of COVID-19 should target major cities and the focus should be on time periods that had shown high risk. Furthermore, the focus should be on elderly and rich people.

Corresponding author
Alfred Ngwira,
alfngwira@yahoo.com

## INTRODUCTION

COVID-19 is a corona virus disease, which was first reported in Wuhan, China, in 2019. It is characterized by severe acute respiratory syndrome (SARS), hence also known as SARS-CoV-2 (*WHO, 2020*). It was declared a pandemic by WHO on 11 March 2020, and it was first reported in Africa, in Egypt on 14 February 2020. The first case in Malawi was reported on 2 April 2020. Since its onset, COVID-19 has been one of the greatest disease pandemics of recent times. It has not only caused a great loss to life, but has also impacted negatively to the global economy, through severe disruption of international travel, tourism and trade (*Pak et al., 2020*). COVID-19 is highly transmissible, but less deadly. For example, compared to human immunodeficiency virus (HIV), as of

29 June 2020, 6 months after COVID-19 emerged in December 2019, there were close to 10 million confirmed cases, and at least 500 thousand deaths worldwide (*Petersen et al., 2020*), compared to 3.3 million new HIV annual cases and 1.8 million deaths during the peak periods in 1997 and 2005 (*GBD 2015 HIV Collaborators, 2016*). Based on the reproductive rate ($R_0$), transmissibility of COVID-19 is also higher than other viruses, where the reproductive rate ($R_0$) of COVID-19 is estimated at 2.5 (1.8–3.6) compared to 2–3 for SARS-CoV and 1918 influenza pandemic, 0.9 for middle east respiratory syndrome coronavirus (MERS-CoV), and 1.5 for 2009 influenza pandemic (*Petersen et al., 2020*).

The spread of COVID-19 is driven by environmental factors like temperature and humidity, where cases spike with low temperature and humidity (*Mecenas et al., 2020*; *Eslami & Jalili, 2020*; *Bherwani et al., 2020*). Demographic and socioeconomic factors like population density, age distribution, income, number of tourists, also play a role in the spread of COVID-19 cases, where high cases are observed with high density, age, income and large number of tourists (*Tantrakarnapa, Bhopdhornangkul & Nakhaapakorn, 2020*). Density has also been found not to be correlated with COVID-19 cases (*Hamidi, Sabouri & Ewing, 2020*), contrary to the belief of many that it is positively correlated with COVID-19 cases. Exposure to air pollution is another possible driver of COVID-19 risk distribution (*Sarmadi et al., 2020*), with increased cases and deaths, where there is high air pollution.

Understanding disease space and time dynamics is important for the epidemiologists as with space distribution, the hot spot areas are marked for intervention (*Snow, 1855*). In addition, possible drivers of the epidemic in those hot spots are suggested for further etiological investigation. Regarding temporal distribution, times with high disease risk are also identified which gives crew to possible causes including, in particular, seasonal changes. A number of studies on spatial temporal distribution of COVID-19 have been conducted (*Chen et al., 2020*; *Yang et al., 2020*; *Gayawan et al., 2020*; *Martellucci et al., 2020*; *Adekunle et al., 2020*; *Arashi et al., 2020*; *Kumar et al., 2020*; *Briz-Redón & Serrano-Aroca, 2020*; *Diop et al., 2020*; *Jia et al., 2020*; *Xie et al., 2020*; *Likassa, 2020*; *Sarfo & Karuppannan, 2020*; *Daw et al., 2020*; *Ye & Hu, 2020*). The majority of these though, have used the geographical information system (GIS) technology as compared to statistical modelling using spatial temporal models. A few studies that have used the statistical approach of spatial temporal analysis to my knowledge are *Gayawan et al. (2020)*, who used the Possion hurdle model to take into account excess zero counts of COVID-19 cases, *Briz-Redón & Serrano-Aroca (2020)*, who used the separable random effects model with structured and unstructured area and time effects, and *Chen et al. (2020)*, who used the inseparable spatial temporal model. In addition, in Africa, spatial temporal analysis of COVID-19 cases has been limited (*Gayawan et al., 2020*; *Arashi et al., 2020*; *Adekunle et al., 2020*). In Malawi, at the time of this study, no study on spatial temporal distribution of COVID-19 cases had been spotted. Only one study that focused on prediction of COVID-19 cases using mathematical models was seen (*Kuunika, 2020*). The aim of this study was to determine the spatial temporal trends of COVID-19

confirmed cases in Malawi while using the spatial temporal statistical models. The objectives of the study were:

- To establish the estimated or predicted risk trend by geographical location;
- To estimate the temporal risk trend of COVID-19 by geographical location.

The article has been organized as follows. First, study methods are described in terms of data collection and statistical analysis. Thereafter, results and discussion of results are presented. Finally, conclusions and implications regarding the findings of the study are made.

## MATERIALS AND METHODS

### Data

The study used the publicly available COVID-19 monthly confirmed cases data for Malawi, which were extracted from the Public Health Institute of Malawi situation reports (https://malawipublichealth.org/index.php/resources/covid-19-sitrep-updates/detail). The study also collected data on demographic and socioeconomic indicators of each district from the Malawi data portal (https://malawi.opendataforafrica.org/lilehtd/covid-19), which were used as independent variables. The following were the demographic and socioeconomic indicators included in the study, whose choice was mainly based on the literature (*Pequeno et al., 2020*; *Hamidi, Sabouri & Ewing, 2020*): total population, population density, percentage of people with running water, percentage of people with no toilet, percentage of people who are aged 65 years and above, poverty incidence and percentage of people with no radio. The population size for each district was used as the expected number of people to be infected in each district. The study period run from April 2, the day the first case was recorded to October 28, the last day in the month of October, the COVID-19 situation report was released. The study period was divided into 6 months as follows: April (2 April 2020–30 April 2020), May (1 May 2020–29 May 2020), June (30 May 2020–30 June 2020), July (1 July 2020–18 July 2020), August (19 July 2020–28 August 2020), September (29 August 2020–30 September 2020) and October (1 October 2020–28 October 2020).

### Statistical analysis

Descriptive analysis involved the time series plot of the cumulative confirmed cases and those who had died of COVID-19 for the whole country from the first month the epidemic began to the month of October 2020. Multiple variable spatial temporal models were then fitted in R using the Bayesian approach while using the integrated nested laplace approximations (INLA). INLA estimation of Bayesian hierarchical models is proved to be faster than the use of Markov Chain Monte Carlo (MCMC) methods. A spatial temporal model assuming linear trend of cases over time as proposed by *Bernardinelli et al. (1995)* was first fitted. That is, let $y_{it}$ be the observed disease cases in region $i$ and at time $t$, then the observed disease cases have the Poisson distribution with the data model defined as $y_{it} \sim Poisson\ (e_{it}r_{it})$, where $e_{it}r_{it}$ is the mean of disease

cases, $e_{it}$ and $r_{it}$ are the expected disease cases and relative risk respectively. The model of the relative risk is then defined as: $\log(r_{it}) = \eta_{it}$, where $\eta_{it}$ is the predictor specified as:

$$\eta_{it} = \mu + u_i + v_i + (\emptyset + \varphi_i). \qquad (1)$$

The model with predictor Eq. (1) will be denoted by A. In Eq. (1), $\mu$ is the overall disease relative risk, $u_i$ is the area level unstructured random effect where area level unit was district, $v_i$ is the area level structured random effect, $\emptyset$ is overall time trend and $\varphi_i$ is the area specific time trend. The area level random effects represent the proxy of unmeasured or unobserved risk factors of COVID-19 in each area, for example, number of lock-downs and average temperature of each area. To incorporate the observed covariates like population density in each area in the model, Eq. (1) can be written as:

$$\eta_{it} = \mu + \omega^T \theta + u_i + v_i + (\emptyset + \varphi_i)t. \qquad (2)$$

In Eq. (2), $\omega$ is a vector of the observed covariates and $\theta$ are the corresponding coefficients. The unstructured area level effects were modelled by the independent normal distribution with zero mean, that is, $u_i \sim N(0, \sigma_u^2)$, and the structured random effects were assigned the intrinsic conditional auto-regressive (ICAR) according to *Besag, York & Mollié (1991)*, that is, $v_i | v_{je\Theta_i} \sim N\left( \dfrac{\sum_{je\Theta_i} v_j w_{ij}}{\sum_{je\Theta_i} w_{ij}}, \dfrac{\sigma_v^2}{\sum_{je\Theta_i} w_{ij}} \right)$. The weakness of model A is the linearity assumption on the effect of time on the relative risk of the disease. To take a more flexible approach on the effect of time, nonlinear spatial temporal models were also explored. The predictor, $\eta_{it}$, or the nonlinear spatial temporal model for the time effect is specified as follows:

$$\eta_{it} = \mu + \omega^T \theta + u_i + v_i + \gamma_t + \beta_t + \delta_{it}. \qquad (3)$$

The model with predictor Eq. (3) will be denoted by B. The $u_i$ and $v_i$ in the model are still the area level unstructured and structured effects respectively as defined in Eq. (1), and $\gamma_t$ and $\beta_t$ are the unstructured and structured temporal effects. The unstructured time effects were modelled by the independent normal distribution with zero mean, that is, $\gamma_t \sim N\left(0, \sigma_\gamma^2\right)$, and the structured temporal effects were assigned the first order random walk prior distribution defined as: $\pi(\beta | \sigma_\beta^2) \propto \exp\left( -\frac{1}{2\sigma_\beta^2} \sum_{t=2}^{T} (\beta_t - \beta_{t-1})^2 \right)$. A second order random walk was also explored in case the data would show a more pronounced linear trend. The second order random walk is defined as: $\pi(\beta | \sigma_\beta^2) \propto \exp\left( -\frac{1}{2\sigma_\beta^2} \sum_{t=2}^{T} (\beta_t - 2\beta_{t-1} + \beta_{t-2})^2 \right)$. The last term in Eq. (3), $\delta_{it}$, represents the interaction between area and time. Four forms of interaction between space and time are possible according to *Knorr-Held (2000)*. The first form of interaction assumes interaction between the unstructured region effect, $u_i$, and the unstructured temporal effect, $\gamma_t$, we denote it by model B 1, and in this case the interaction effect is assigned the independent normal distribution, that is, $\delta_{it} \sim N(0, \sigma_\delta^2)$. The second type of interaction, is the interaction of structured area effect, $v_i$, and the unstructured temporal effect, $\gamma_t$,

we denote it by model B 2. This form of interaction assumes conditional intrinsic (CAR) distribution for the areas for each time independently from all the other times. The third is the interaction between unstructured area random effect, $u_i$, and the structured temporal effect, $\beta_t$, we denote this by model B 3. The prior distribution for each area is assumed to be a second order random walk across time. The last possible space time interaction is that between area structured effect, $v_i$, and the structured time effect, $\beta_j$, we call it model B 4. In this case, the second order random walk prior that depends on neighboring areas was assigned for each area. The prior for the variance parameters was the log-Gamma (1, 0.0005), which is considered as the default minimal informative prior (*Blangiardo et al., 2013*). The fixed effects were assigned the default normal, that is, N (0, 0). The model choice was by the deviance information criteria (DIC) as proposed by *Spiegelhalter et al. (2002)*, where a smaller DIC means a better model in terms of fit and complexity. It is the sum of the measure of model fit, called the deviance, denoted by $\bar{D}$ and the effective number of parameters denoted by $P_{\bar{D}}$. Since the fitted model was fully Bayesian, the selected model was checked for sensitivity by checking the sensitivity of the hyper-parameters for changes in their prior distributions (*Roos & Held, 2011*). The log-Gamma (1, 0.01) and log-Gamma (0.5, 0.001) priors were the alternative priors in the sensitivity analysis of the hyper-parameters. The selected model was then used to estimate the relative risk, $r_{it}$.

## RESULTS

Figure 1 shows the graph of cumulative confirmed and dead cases of COVID-19 from the time the first case was reported, 2 April 2020–28 October 2020. There were 5,904 confirmed cases and 184 deceased cases as of 28 October 2020. Generally, there were low total cases of those who had died of COVID-19 as compared to those who were confirmed.

Table 1 shows the DIC of the fitted spatial temporal models defined in the methods section. Model B 11 had smallest DIC than the rest of the models, though it was not significantly different from model B 12, since the DIC difference was not greater than 3 (*Spiegelhalter et al., 2002*). Sensitivity analysis of model B11 showed that the hyper-parameters posterior distributions were generally stable for small changes in their prior distributions. For example, Fig. 2 shows similar posterior distributions of the variance of unstructured effect of area, and variance of interaction effect of area and time for different prior distributions: log-Gamma (1, 0.0005), log-Gamma (1, 0.01) and log-Gamma (0.5, 0.001). The first two priors resulted into posteriors which were almost super-imposing, that is, one laid over the other. The results of model B11 are therefore presented and discussed. Table 2 presents results of model B 11. The proportion of elderly people (65+ years) was positively associated with COVID-19 ($\beta = 1.272$, 95 % CI [0.171, 2.370]). The positive coefficient of elderly people means that high proportion of elderly people compared to low proportion of elderly people is associated with increased risk of COVID-19. Poverty incidence was negatively associated with COVID-19 risk ($\beta = -0.100$, 95% CI [−0.136, −0.065]). Negative coefficient of poverty incidence means that high poverty incidence is associated with reduced risk of being diagnosed with COVID-19 as compared to low poverty incidence. All other sociodemographic factors

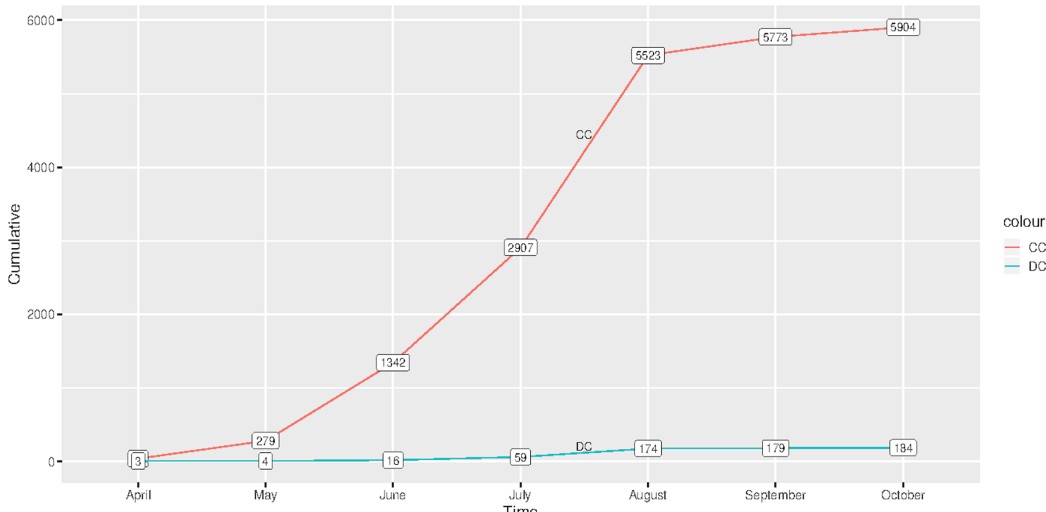

**Figure 1 Cumulative cases of confirmed and those who had died of COVID-19 in Malawi from 2 April 2020 to 28 October 2020.** (CC) Confirmed cases and (DC) Deceased cases.

**Table 1 DIC of the fitted models.**

| Model code | Description | DIC |
|---|---|---|
| A | Parametric linear time trend | 13,675.03 |
| B 11 | Non parametric time trend, type 1 interaction, RW2. | 1,052.041 |
| B 12 | Non parametric time trend, type 1 interaction, RW1. | 1,052.071 |
| B 21 | Non parametric time trend, type 2 interaction, RW2. | 1,094.861 |
| B 22 | Non parametric time trend, type 2 interaction, RW1. | 1,094.922 |
| B 31 | Non parametric time trend, type 3 interaction, RW2. | 1,107.471 |
| B 32 | Non parametric time trend, type 3 interaction, RW1. | 1,081.685 |
| B 41 | Non parametric time trend, type 4 interaction, RW2. | 1,098.379 |
| B 42 | Non parametric time trend, type 4 interaction, RW1. | 1,073.298 |

were not significant determinants of COVID-19 confirmed cases, since the p-values were greater than 0.05. All the random effects, that is, the effect of time, the effect of location, including the interaction effect of location and time, were significant predictors of the risk of contracting COVID-19, since the estimated variance parameters were significantly greater than zero, considering that the confidence intervals excluded zero. Area level spatial effects and the effects of time, modelled by the second order random walk prior (RW2), were highly significant as evidenced from the large variance parameters.

The spatial temporal distribution of overall fitted risk (Figs. 3 and 4), shows that by space (Fig. 3), areas in and around the cities like Mzimba in the northern region, Lilongwe in the center and Blantyre in the south, were at increased risk of being confirmed of COVID-19 compared to the rest of the areas. Over time, from April to October (Fig. 4), the

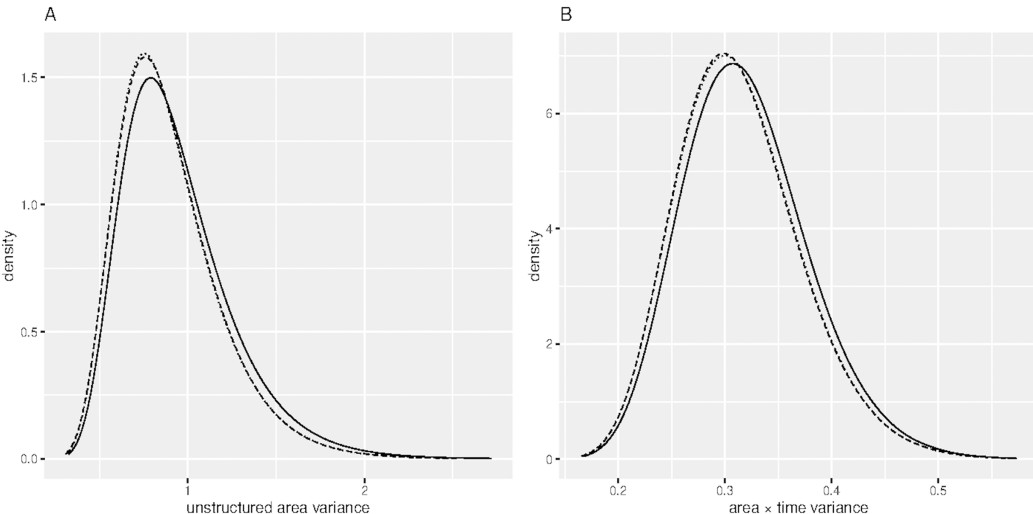

**Figure 2 Sensitivity analysis of the hyper-parameters.** (A) Posterior distribution of unstructured area variance and (B) Posterior distribution of area × time variance. (Dashed) log-Gamma (1, 0.0005), (Dotted) log-Gamma (1, 0.01) and (Solid) log-Gamma (0.5, 0.001).

**Table 2 Spatial temporal model of COVID-19 monthly confirmed cases (model B 11).**

| Variable | Coefficient (95% confidence interval) |
| --- | --- |
| Fixed effects | |
| Population density | −0.003 [−0.007, 0.001] |
| Proportion of households with no toilet | 0.089 [−0.071, 0.250] |
| Proportion of households with running water | −0.020 [−0.073, 0.033] |
| Proportion of households with no radio | −0.095 [−0.216, 0.025] |
| Proportion of people with age 65+ years | 1.272 [0.171, 2.370] |
| Poverty incidence | −0.100 [−0.136, −0.065] |
| Variance parameters | |
| Area (i.i.d) | 1.240 [0.621, 2.170] |
| Area (spatial) | 1,870.72 [128.861, 6,839.73] |
| Time (i.i.d) | 108.64 [0.331, 689.52] |
| Time (RW2) | 1.45 [0.020, 8.29] |
| Area x Time (i.i.d) | 3.30 [2.252, 4.68] |

risk in all districts was increasing from April to August, where it started to decline to October. Blantyre was consistently at high risk to COVID-19 over time compared to other areas. Most of the districts in the rural areas, were consistently at low risk of contracting COVID-19 over time compared to urban districts. Figure 5 presents the spatial risk of contracting COVID-19. Spatial risk represents the residual risk due to unobserved or unmeasured factors of COVID-19. In general, the spatial risk looks high in many areas in June, July, August and a little bit high in September compared to April, May and October.

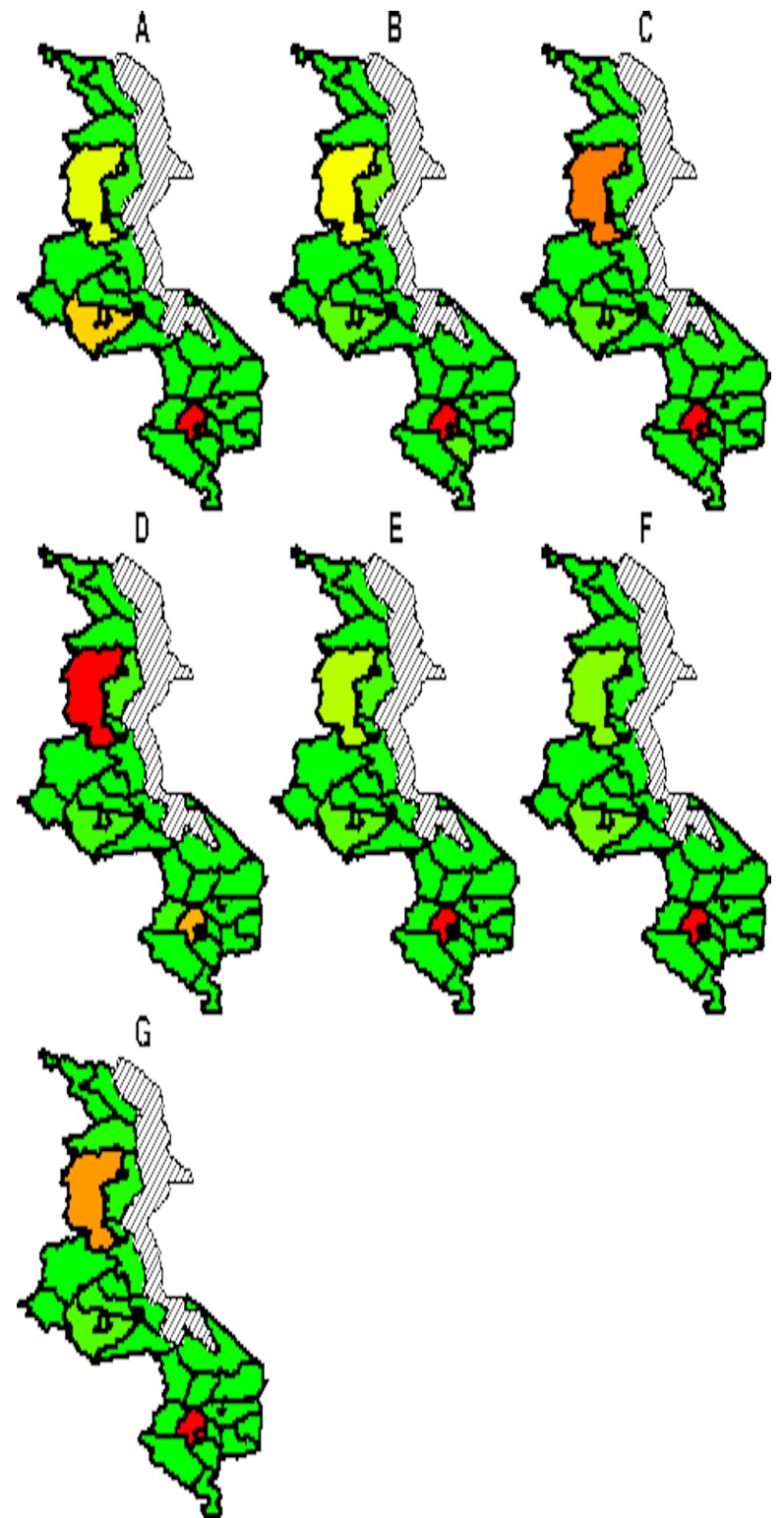

**Figure 3 Evolution of overall predicted risk by district over time.** (A) April, (B) May, (C) June, (D) July, (E) August, (F) September and (G) October. Green (low risk), yellow (medium risk) and red (high risk).

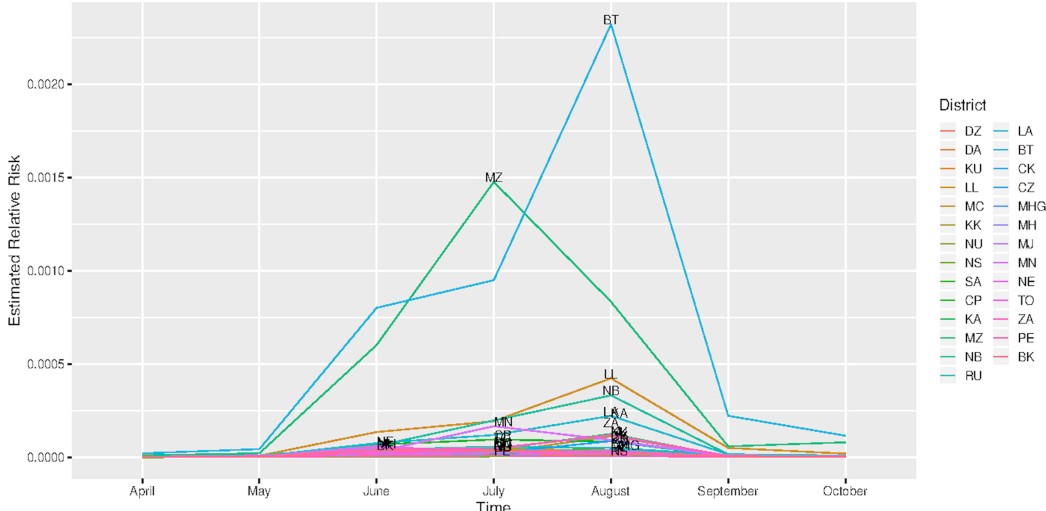

**Figure 4 Evolution of relative risk by district over time.** (DZ) Dedza, (DA) Dowa, (KU) Kasungu, (LL) Lilongwe, (MC) Mchinji, (KK) Nkhota-Kota, (NU) Ntcheu, (NS) Ntchisi, (SA) Salima, (CP) Chitipa, (KA) Karonga, (MZ), Mzimba/Mzuzu, (CK) Chikwawa, (CZ) Chiradzulu, (MHG) Machinga, (MH) Mangochi, (MJ) Mulanje, (MN) Mwanza, (NE) Nsanje, (TO) Thyolo, (ZA) Zomba, (PE) Phalombe and (BK) Balaka.

## DISCUSSION

The study looked at spatial temporal distribution of COVID-19 in Malawi from the time the first case was recorded, 2 April 2020–28 October 2020, while using the inseparable statistical spatial temporal model. The use of inseparable model, allowed the investigation of the joint or interaction effect of time and location on the risk of being confirmed of COVID-19. The use of nonparametric model for time effect (RW2), also enabled the capturing of the subtle influences of time on the risk of contracting COVID-19.

The study finds significant effect of poverty incidence and proportion of elderly (65+ years) people on COVID-19 risk. The results on effect of poverty on COVID-19 risk are consistent with the literature (*Tantrakarnapa, Bhopdhornangkul & Nakhaapakorn, 2020*; *Sarmadi et al., 2020*), where it was found that high income compared to low income was positively associated with COVID-19 cases. Low risk of COVID-19 among poor people than rich people may be due to the fact that such people are less likely to travel to cities and abroad where they can be exposed to the disease. In addition, low COVID-19 risk among poor people may be due to the fact that in Malawi and in poor countries in general, disease monitoring or surveillance inform of testing is very minimal, which results in less people to be diagnosed of the disease. The higher risk of COVID-19 for higher income than lower income entails the same relationship between age and COVID-19 since wealth is a proxy for age as wealth is accumulated over time. The increased risk of COVID-19 for elderly people than young people is due to the fact that elderly people have a weaker immune system than the youths and hence, they are susceptible to infection. Of special interest, the results on population density show negative correlation with COVID-19 infection rate and that the association is not significant. This is contrary to the common understanding that a virus is positively associated with density. Of course, this is

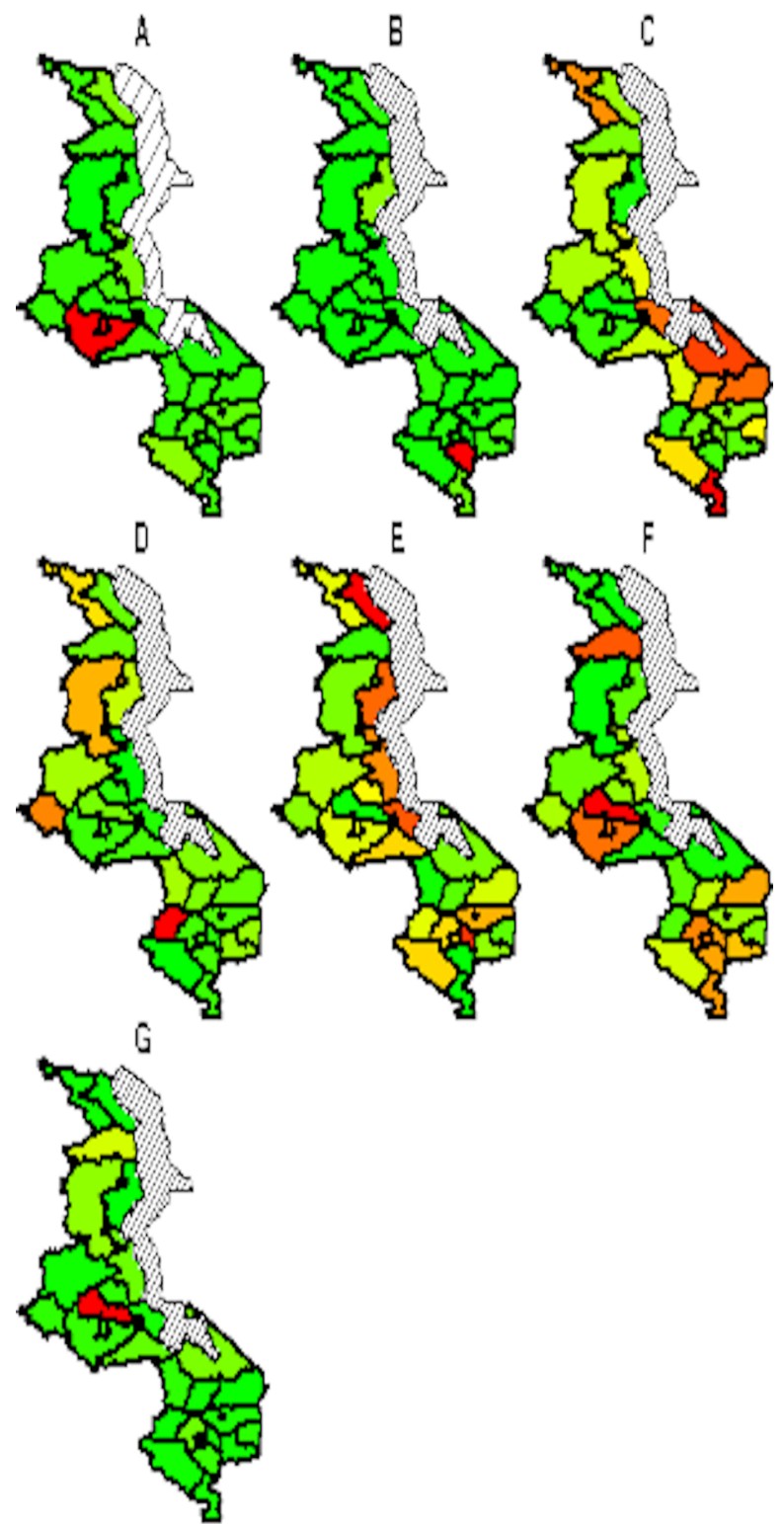

**Figure 5 Evolution of spatial risk of COVID-19 over time (Interaction effect of location and time on exponential scale).** (A) April, (B) May, (C) June, (D) July, (E) August, (F) September and (G) October. Green (low risk), yellow (medium risk) and red (high risk).

not strange, considering that previous studies have found mixed results, with some finding negative association (*Carozzi, Provenzano & Roth, 2020*), and others finding positive association (*Hamidi, Sabouri & Ewing, 2020*; *Pequeno et al., 2020*; *Sarmadi et al., 2020*), and that in many cases, there has been no significant association. It is argued that the negative effect of density on COVID-19 infection rate or lack of significant positive association may be attributed to some mediating factors that offset the usual positive association. For example, high density areas may attract young folks who are usually resistant to COVID-19 infection. In addition, high density populations are positively correlated with proxies of social distancing (*Carozzi, Provenzano & Roth, 2020*) due to the increased consciousness of being at risk, which in turn reduces COVID-19 cases.

The spatial distribution of COVID-19 risk in Malawi at the given time period, shows the cities and the surrounding areas being at increased risk than the rural areas. The explanation to the observed spatial gradient is a matter of conjecture. One possible factor driving the observed spatial pattern would be the population size and population density. The cities have higher population density than the rural, and COVID-19 is therefore more likely to spread fast through the movement and frequent contact between people. Case comparison investigations have found positive correlation of population density and COVID-19 (*Penerliev & Petkov, 2020*), where for example, in Italy, Lombardy, the population density is three times higher than in Piedmont and the incidence rate was also over three times higher than in Piedmont. Evidence of high population density as a risk factor of disease transmission has been seen in India, where influenza transmission rates have been found to increase above a population density of 282 people per square kilometer (*African Centre for Strategic Studies, 2020*).

The other possible contributor to the observed rural city spatial gradient of COVID-19 risk in Malawi, would be international exposure. In this case, cities have higher international exposure than the rural through international flights among others, which would mean more imported cases. Evidence of international exposure as a risk factor of COVID-19 transmission has been observed in Africa as a whole, where countries with high international exposure like South Africa, Nigeria, Morocco, Egypt, and Algeria have had higher COVID-19 cases than their counterparts. International exposure as a fuel of COVID-19 transmission has also been documented in Brazil where it was found that cases increased with increase in international flights jetting into the country (*Pequeno et al., 2020*). The ability to testing may be another catalyst of COVID-19 risk distribution in Malawi, where in the urban centers, a large number of people are tested, and hence, more people are reported to have COVID-19 than in the rural setting. COVID-19 testing rate has been confirmed to be positively correlated with virus infection rate (*Hamidi, Sabouri & Ewing, 2020*).

Regarding the temporal distribution, a spike of the disease risk from June to August, might be attributed to the effects of unrestricted presidential election political rallies which were being held in most cases, in urban centers. The presidential election was held on 23 June 2020. In addition, the rise in COVID-19 cases during this time would be attributed to the decreasing temperatures at this time of the year as this is the time of cold season. The decline of disease risk from August to October may be due to the increasing

temperatures, as this time, marks the beginning of hot season. Negative correlation between COVID-19 cases and temperature has been documented (*Pequeno et al., 2020*).

The study did not go without weaknesses. The first weakness was that, due to the absence of population size for each area at each time point, the base population at risk for each area was assumed to be constant across time which was not practically valid. The other weakness was that the study did not take into account the effect of other important factors of COVID-19 like the number of lock-downs and average temperature of each district, while investigating the spatial temporal distribution of the risk. Nevertheless, the results of the study are still valid, as all unmeasured factors were captured through the area level and time random effects which were considered as proxies for such factors.

## CONCLUSION

The study found a significant effect of both location and time on COVID-19 risk, and the effect of either of the two depended on the other, that is, interaction. The risk of COVID-19 for major cities and the surrounding areas was high compared to the rural districts and that over time, the risk for rural areas remained relatively lower than in cities. The risk of contracting COVID-19 was increasing from June to August and started to decline thereafter. Elderly people were positively correlated with COVID-19 infection rate, and poverty incidence had a negative effect on COVID-19 risk. The implications of the study are that future interventions to halt the disease transmission, should target the major cities like Blantyre, Zomba, Mangochi, Lilongwe and Mzuzu, and that by time, attention should be paid to the month of June, July and partly August, when it is very cold. Furthermore, attention should be paid to elderly people and rich people.

### Funding
The authors received no funding for this work.

### Competing Interests
The authors declare that they have no competing interests.

### Author Contributions
- Alfred Ngwira conceived and designed the experiments, performed the experiments, analyzed the data, prepared figures and/or tables, authored or reviewed drafts of the paper, and approved the final draft.
- Felix Kumwenda conceived and designed the experiments, performed the experiments, prepared figures and/or tables, and approved the final draft.
- Eddons C.S. Munthali analyzed the data, authored or reviewed drafts of the paper, and approved the final draft.
- Duncan Nkolokosa conceived and designed the experiments, prepared figures and/or tables, authored or reviewed drafts of the paper, and approved the final draft.

## Data Availability

The data and the R code are available in the Supplemental Files.

## Supplemental Information

Supplemental information for this article can be found online at http://dx.doi.org/10.7717/peerj.11003#supplemental-information.

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
