# Peer review of "Spatial temporal distribution of COVID-19 risk during the early phase of the pandemic in Malawi"

_PeerJ, doi:10.7717/peerj.11003_

## Round 0.1 · original submission · Major Revisions

· Academic Editor

Major Revisions

I agree with most comments made by the reviewers. Please pay particular attention to comments regarding professional English writing, the lack of covariates declared in the analyses, and extra effort to make your analyses reproducible.

One concerning issue is the limited data and results you report. Could you improve on the 6-week time period under study? As stated by one reviewer, considering the early epidemic in Malawi would be valuable to the study. Please seriously consider these suggestions.

·

Basic reporting

Abstract:

"Future or present strategies to avert
the spread of COVID-19 should target major cities by limiting international exposure."

You do not explicitly study the effect of international exposure, so this sentence
does not belong in the abstract.


Introduction and references:

Line 316: In the reference for "(WHO, 2020)", on page 15, the content of the
URL following the ? is unnecessary:
"https://www.who.int/docs/default-source/coronaviruse/situation-reports/20200810-covid-19-sitrep-203.pdf"
is sufficient.

Line 33-34: "Since its onset, COVID-19 has been one of the greatest disease
pandemics of all times."

This may be true, but there is no reference supporting it. How do the
incidence and prevalence compare with polio, influenza, and HIV/AIDS? I would
like to see a short comparison with two other pandemics at their peaks.
This should demonstrate that COVID-19 is similar, and emphasise its
seriousness.

I would also (optionally) change this sentence to:
"Since its onset, COVID-19 has become one of the greatest pandemics of modern
times."

Line 34: The statement:
"From its discovery in December in China, 19 718 030 people world over..."

has several problems, most of which are the fault of the WHO and not the
authors. For example, the statistic is implausibly precise. The uncertainties in
monitoring and surveillance of COVID-19 are very high: it is meaningless to
quote statistics to seven decimal places, unless the uncertainties in the
measurement process are less than one part in ten million. This is certainly
not the case with the current pandemic.

This figure is also changing daily. It was already September when I received
this paper for review, and will be October -- at the earliest -- when this
paper is published. Rather than emphasising the relevance of the paper, I feel
this sentence ties it to a specific time, and may cause it to age poorly.
However, the author's methodology is fundamentally sound, and I believe it is
a valuable contribution.

My recommendation is simply to delete this sentence, and provide more detail
on historical context elsewhere.

Lines 43-44: "Understanding disease space and time dynamics is important for
the epidemiologists as with space distribution, the hot spot areas are marked
for intervention."

I agree. A reference to John Snow's 1854 study of a cholera outbreak in London
would help to emphasise the importance of spatio-temporal studies to our
understanding of disease, and to suggest policy interventions. (John Snow, On
the Mode of Communication of Cholera, 1855. Available from the UCLA Fielding
School of Public Health here: http://www.ph.ucla.edu/epi/snow/snowbook.html)

This is an optional edit but I think it would add helpful historical context.

The rest of the introduction is good.

Experimental design

Statistical analysis:

Overall, this section and the design are well-described. I would like to
commend the authors for providing both their original data and the R code used
for the analysis.

Lines 94-96:
"The potential covariates would be selected for further multiple variable modelling if
their p-values were less than 0.20."

Why 0.20? Please explain why you think this was an appropriate threshold.
(Around one or two paragraphs.)

Validity of the findings

Figures:

Figures 1-3:

The colour scheme for all figures is unclear, and the resolution is somewhat
low. Please add point markers to these plots and legends.

The key for figure 3 is unclear. Are the labels on the right in roughly the
same order as the lines on the left, from top to bottom? I can't tell which
curve is which. This would be solved by adding distinct point markers to the
plot and legend.

Figure 4 looks great. In the text, you mention the possible effect of
political rallies during the 2020 presidential election. Would it be possible
for you to add markers to the maps in figure 4 that show when and where the largest political rallies happened during this period?

Optional improvements to all figures:

Ideally, I would like to be able to reproduce these figures using the data in the
supplemental material, and an R or Python program. Can this be incorporated in
your supplemental material?

The resolution of several figures is quite low. Is it possible for you to
upload vectorised SVG, postscript, or PDF versions, either in the main paper
or in the supplemental material?

Supplemental material and R code:

Dependencies:

The packages INLA, spdep, ggplot2, and BayesX are not installed by default,
but are available in CRAN. To install these on Debian GNU/Linux, I did the
following:

apt install r-cran-surveillance r-cran-spdata r-cran-spdep r-cran-ggplot2

I also had to download and install the BayesX library by starting an R prompt
and entering:

> install.packages("BayesX")

A short README.txt containing the above commands would be very helpful.

Line 5: The file "malawi.bnd" was not included the supplemental material, and
I was unable to find it online. Without this file, I was unable to execute the
notebook.

Line 10: When downloaded, the file "covid19data.csv" will not be located at
"C:/p/covid19data.csv". I found that a simple fix was to change this to a
relative path:

d<-read.csv("./covid19data.csv")

If you can confirm that this also works on your machine I think this should be
changed.

Additional comments

General comments and summary:

I enjoyed reading this report on spatio-temporal dynamics of COVID-19 risk in
Malawi. I was pleased to find it to be higher quality than many similar papers
on COVID-19 that have been forwarded to me, with a well-described methods
section, and basically sound conclusions. I have recommended several minor
revisions to improve the context and reproducibility of the work before it can
be accepted for publication.

I am especially impressed by the effort the authors have put into ensuring
their work is reproducible, and was able to test their R code on my
machine. After the last issue relating to the provided files is resolved, I
will be very happy with the quality of the supplemental material.

There are some sentences in the abstract and introduction, mainly regarding
historical context, that I think should be rephrased. Importantly, there is
one methodological choice -- setting significance threshold P=0.20 -- that
needs better explanation.

I have also suggested several improvements to all figures and the supplemental
material.

Reviewer 2 ·

Basic reporting

This study aims to estimate/predict spatiotemporal changes in COVID-19 infected risk in Malawi. I must be honest that the manuscript is far from well-written and has low-quality figures.

Experimental design

Because there are many published works with a similar topic worldwide, this study's originality is limited. In my opinion, research questions were not attractive. If the author could compare different risk prediction methods or find the main factors affecting the risk distribution, the manuscript will be much more meaningful. The p-value should not be used because it can be misleading. Please check "Psychology journal bans P values" [doi:10.1038/519009f] and newer related publications.

Validity of the findings

The main result is "the spatial distribution of COVID-19 showed major cities being at greater risk than rural areas ..." and recommend that "... avert the spread of COVID-19 should target major cities by limiting international exposure" are not new things. Because there is no model validation in this study, how the prediction could be applied for a future event is still a question.

Reviewer 3 ·

Basic reporting

Dear authors,

Thanks for the opportunity to review your manuscript.

Although the COVID pandemic would require a global effort to promote high quality investigations so that to improve our understanding and design adequate control strategies, I consider you work as deficient in the writing, data/information and approach.
At first, English needs to be improved throughout the ms. There are sentences really hard to follow and the language does not appear to follow scientific terminologies.

Secondly, you are using a very short period of time, 6 weeks, to perform spatial temporal analysis. However, in the analysis you omit almost three months of data since the epidemic started (or first index case) on April 2nd. The first stage of the epidemic is crucial for the understanding of the overall results, particularly the early spread between districts. Not knowing such info, it is really hard to hypothesized about the spread of the virus in Malawi.

Third, you reported very limited covariates in your analysis (pop. size, density and water data). Moreover, none of them showed association to COVID cases, So, in addition to not justify their selection and exclusion of other putative risk factors that may be available, the results are not relevant at all.

Overall, I congratulate the approach used but the very limited data and results are insufficient to consider this work as a publication in this journal.

Experimental design

no comment

Validity of the findings

no comment

Additional comments

no comment

---

## Round 0.2 · Major Revisions

· Academic Editor

Major Revisions

Dear authors,

While the reviewers agree that you have improved the manuscript to some degree, they still require more substantial changes. All of these concerns need to be taken care of. Please, address these comments and submit a revised version.

·

Basic reporting

No comment

Experimental design

Thank you for including the map of Malawi. I still required some changes to the attached R code to get it to run, and needed to install R-INLA from here (https://www.r-inla.org/download-install). I could reproduce figures 1 and 3 but not figures 2 or 4.

Validity of the findings

I was very surprised by the finding of negative association of incidence with poverty and population density. Have the authors ruled out differences in monitoring/surveillance, or a correlation between wealth and age? I can imagine that lower access to healthcare would result in less effective monitoring: if so, we might not be able to trust data from poorer regions. I can also imagine that older people are generally wealthier as they have had a longer time to accumulate savings: if so, association with wealth might simply be a proxy for association with age.

To address this, I think a few sentences addressing the above points are necessary.

Reviewer 2 ·

Basic reporting

The manuscript is improved, but my concern about the study's originality still remains.

Experimental design

(1) The authors compare their work with that of Schrödle and Held (2010), which does not make sense for me. While Schrödle and Held mainly focused on the mathematical aspect of how INLA can be applied for spatiotemporal disease mapping, this manuscript is a case study of INLA's application. Therefore, I would like to see more about the model validation or what this case study can contribute to.

(2) In my opinion, this work is not attractive if the authors only focus on investigating the distribution of COVID-19 in both space and time in Malawi. By this limitation, the manuscript should be submitted to a regional journal. I repeat my comment from the first review round again. If the author could compare different risk prediction methods or find the main factors affecting the risk distribution, the manuscript will be much more meaningful. Some results of driven factors are shown in the revised manuscript. Why not do a deeper analysis (using additional statistic methods) and discuss in more detail those factors? We need a solid analysis that can help to identify the main driven factors.

(3) Banning P-value's usage is a wide discussion in the scientific community. It is naive if you argue this point by referring to an opinion on Figshare.

Validity of the findings

Minor points:
- Figures 2 and 4: there are no legends. The distance between maps is too wide.
- Figure 3: I cannot read the text inside the plot. Why not increase the height of the plot?
- Some typos: "COVI-19" in lines 50 and 260.

Reviewer 3 ·

Basic reporting

Thanks again for the opportunity to review this ms. I can see authors did an effort to improve English grammar and update methods and data, however, I still think the paper needs major revisions.

I do not think the keywords are in line with the work. I would suggest spatial epidemiology, spatial risk, COVID spread, spatiotemporal modeling, etc. I do not think "Inseparable, separable" would be useful keywords.

Please, make an attempt to use colors for the spatial figures. Maybe consider green (low risk), yellow (medium risk) and red (high risk) for reporting. Black and white make really difficult to capture risk.

L 14 - what is the epidemiological spatial unit? District, province, region, commune, etc? Please mention it.

L 21 - "Proportion of elderly people was positively associated with COVID-19 risk than what???? - remember in epidemiology you compare populations on risk. Please, be specific about this throughout the ms.

L 23 - (CI = -0.136 ; -0.065). - Maybe using ; would be better than -.

L 24 - "to avert" - maybe "to limit".

L 40-42: What is the meaning of that sentence? Is that SARS-CoV-2 has the highest average R0? is more transmissible? Please, state.

L 43 - is the distribution or the spread driven by?

L 48 - i do not see a connection between density and super-spreading. The latter is an individual pattern, well recognized in the COVID pandemic, nothing to do with the density.

Experimental design

Overall, nothing is mentioned in the paper about non-pharmaceutical interventions such as movement restrictions at local or national scales (i.e., lockdowns). How about their role in limiting the spread in the country? Is that accounted for? Please, describe.

Validity of the findings

Little is mention about the relative risk reported results. A risk ratio (RR), also called relative risk, compares the risk of a health event (here COVID incidence) among one group with the risk among another group. A RR of 1.0 indicates identical risk among the two groups. A risk ratio greater than 1.0 indicates an increased risk for the group in the numerator, usually the exposed group.RR < 1.0 indicates a decreased risk for the exposed group, indicating that perhaps exposure actually protects against COVID occurrence. Based on Fig. 3, your RR goes from 0 to ~0.003. This is very strange reporting RR and I would suggest you need to discuss this with an epidemiologist to interpret such results properly.

Additional comments

Although authors did an attempt to update data and improve English in many ways, I believe you need to discuss results with an epidemiologist to properly interpret results and some key concepts.

---

## Round 0.3 · Minor Revisions

· Academic Editor

Minor Revisions

Two reviewers have submitted their assessment of your manuscript. Please, address all of their comments including the resolution of the images, and submit a revised version. I will make a decision after you resubmit.

·

Basic reporting

Looks fine

Experimental design

After installing the relevant dependencies the R code produced several graphs, but returned some errors.

I noticed some typos in the R code again: "formula33" in code.R should be "formula32" I think. Also "formula44" should be "formula42". After these corrections were made, i got the error:

Error in plot.new() : figure margins too large
Calls: drawmap -> plot -> plot.default -> plot.new
Execution halted

Also, I am concerned that the authors have not conducted a sensitivity analysis of their fitting parameters. This would give a clearer picture of how robust the findings are, and also serve to validate the model. I can suggest bootstrapping confidence intervals, which I know to be possible in R, but other methods are possible.

Validity of the findings

Looks good, but issues with the supplemental code and parameter sensitivity/model validation should be addressed.

Reviewer 2 ·

Basic reporting

Because this manuscript is a case study that applies an available model for investigating the distribution of COVID-19, its originality is limited. By this limitation, the manuscript should be submitted to a regional journal. Although the authors try to improve some parts of the manuscript, the figure quality is still low, and the findings are not attractive.

Experimental design

No comment.

Validity of the findings

No comment.

---

## Round 0.4 · accepted · Accept

· Academic Editor

Accept

I have considered the changes made and I'm moved to accept the manuscript. Congratulations.